# Early Recovery of Left Ventricular Function After Revascularization in Acute Coronary Syndrome

**DOI:** 10.3390/jcm9010024

**Published:** 2019-12-20

**Authors:** Rafik Shenouda, Ibadete Bytyçi, Mohamed Sobhy, Michael Y. Henein

**Affiliations:** 1Institute of Public Health and Clinical Medicine, Umea University, 90187 Umea, Sweden; 2International Cardiac Centre—ICC and Alexandria University, Alexandria 21500, Egypt; 3Cardiology Department, Faculty of Medicine, Alexandria University, Alexandria 21500, Egypt

**Keywords:** acute coronary syndrome, deformation parameters, left ventricular function

## Abstract

The aim of this study was to assess the accuracy of echocardiographic techniques in detecting the early recovery of left ventricular (LV) function after revascularization in acute coronary syndrome (ACS). In 80 consecutive patients with ACS (age 55.7 ± 9.4 years, 77% male, 15% with CCS Angina III), an echocardiographic examination of left ventricle regional wall motion abnormalities (LV RWMA), peak systolic strain rate (PSSR), peak systolic strain (PSS) and end systolic strain (ESS) was performed before and after percutaneous intervention (PCI). Of the 80 patients, one vessel stenosis (>70%) was present in 53 (66%), two vessel disease in 12 (15%) and multivessel disease in 15 patients (19%). In total, 51% of patients had hypertension, 40% diabetes and 23% dyslipidemia. After PCI, regional PSS, ESS and PSSR of their segments subtended by the culprit vessel improved; left anterior descending-LAD, circumflex-LCx and right coronary-RCA (p<0.05 for all) as well as global S and SR (*p* < 0.05 for all). In univariate analysis, hypertension (HTN) (β = −0.294 (−0.313–0.047), *p* = 0.009, smoking β = −0.244 (−0.289–0.015) =0.03, WMA β = −0.317 (−0.284–0.014), *p* = 0.004 and the number of diseased vessels β = −0.256 (−0.188– 0.054) p=0.03 were predictors of delta global SR. In multivariate analysis, only HTN β = 0.263 (0.005–3.159) and the number of diseased vessels β =0.263 (0.005 - 3.159), p=0.04) predicted delta global SR. In ACS, the echocardiographic regional myocardial deformation is accurate in detecting early recovery of LV myocardial function after culprit lesion revascularization. Also, the findings of this study support the current practice regarding the crucial importance of proximal epicardial vessel PCI treatment on LV function compared to more distal lesions.

## 1. Introduction

Acute coronary syndrome (ACS) remains a major cause of death and morbidity worldwide [1], despite the well-established treatment strategies including direct revascularization with percutaneous coronary intervention (PCI) [2]. The recent significant improvement in PCI is related to the implantation of various types of drug-eluting stents and the use of dual antiplatelet therapies; however, patients with ACS still have a higher risk for cardiovascular events compared to those with no acute events [3].

The evaluation of cardiac function, in particular left ventricular (LV) function before PCI, adds to the accurate design of treatment strategy. Recently, beyond the assessment of ejection fraction (EF) as a global marker of LV function, the evaluation of myocardial deformation by regional or global strain and strain rate (SR) has been introduced in routine clinical practice [4] and in acute cardiac syndromes. Global and regional myocardial deformation assessment by echocardiography is a current non-invasive ultrasound imaging technique that allows for the evaluation of global and regional LV myocardial function [5,6] and their early response to treatment.

The aim of this study was to assess the early effect of coronary revascularization by PCI for ACS on the recovery of segmental and regional LV function as well as the impact of the proximity of the targeted lesion on cavity function recovery.

## 2. Materials and Methods

### 2.1. Study Population

Eighty consecutive patients presenting to the emergency room of the International Cardiac Centre (ICC), Alexandria, Egypt from January 2018 to October 2018 were included in this study. ACS was defined and patients were managed according to the European Guidelines [7]. Patients with acute chest pain and persistent (>20 min) ST-segment elevation, i.e., STEMI (50 patients) generally reflected an acute total coronary occlusion. All patients received immediate myocardial reperfusion treatment by primary coronary intervention and angioplasty or fibrinolytic therapy, as appropriate. Patients with acute chest pain but no persistent ST-segment elevation, referred to as Non-STEMI (30 patients), (Table 1) included transient ST-segment elevation, persistent or transient ST-segment depression, T-wave inversion, flat T waves or pseudo-normalized T waves or completely normal ECG at admission. 

### 2.2. Exclusion Criteria

All patients with previous myocardial infarction, PCI, coronary artery bypass graft surgery (CABG), more than mild valve disease, atrial fibrillation on admission, LV ejection fraction (EF) <35% or hemodynamic instability were excluded from the study. 

### 2.3. Study Protocol 

On admission, all patients had a preliminary ECG, a blood sample withdrawn for troponin and creatine kinase-MB (CK-MB) in addition to other routine laboratory investigations including hematology and biochemistry. 

### 2.4. Echocardiographic Imaging

The patients also underwent a full echocardiographic examination by the clinician on call, including tissue Doppler imaging (TDI) (first 30 patients) and speckle tracking echocardiography (STE) (the remaining 50 patients). The echocardiographic examination was performed using a Philips system equipped with an adult 1.5–4.3 MHz phased array transducer. The standard views from the parasternal long and short axis and the apical four-chamber views were acquired. The LV end-diastolic dimension (LVDd) and end-systolic dimension (LVDs) were measured in the parasternal LV long-axis view, and the fractional shortening was calculated. The interventricular septal thickness and LV posterior wall thickness were measured at the end-diastole and end-systole in the LV short-axis view at the basal level. The LV EF was calculated using the biplane method [8]. 

The tissue Doppler segmental myocardial velocities were measured at the basal segments of the longitudinal segments according to the conventional guidelines of the American Society of Echocardiography and European Association of Cardiovascular Imaging [9]. All Doppler echocardiographic recordings were made at a sweep speed of 50–100 mm/s^−1^ with a superimposed ECG (lead II), and the measurements were taken as means of three consecutive cardiac cycles. Speckle tracking images were obtained with the patient in the left lateral decubitus position, from the parasternal short axis (mitral leaflet, annular and apex level) and from apical 4-, 3- and 2-chamber views, at rest. Two-dimensional grey scale images were obtained at a frame rate of 70–80 Hz during three cardiac cycles and were then digitally stored for off-line analysis. The off-line analysis of the TDI and STE cine loops was performed by two investigators using commercially available software (General Electric, EchoPac version BT 13,113.0, Waukesha, Wisconsin, USA). After manually outlining a clear myocardial border at the end-systole, the region of interest (ROI), generated automatically by the software, was manually adjusted and the peak systolic strain (PSS), peak systolic strain rate (PSSR), and peak end systolic strain rate (PESSR) at the basal, middle, and apical levels were all measured [10]. 

The strain analysis through the entire cardiac cycle provided continuous values of global and segmental strain for all segments simultaneously. The regional strain was calculated by combining the values of the segments subtended by each of the three main epicardial coronary arteries, the left anterior descending (LAD), circumflex (LCx) and right coronary artery (RCA). The regional values for the basal and apical segments were also combined based on the respective arterial blood supply. In addition, the improvement of myocardial deformation after PCI was calculated as delta strain.

According to clinical presentation, the patients underwent either a primary PCI (for STEMI) or elective PCI within the first 48 hours of admission (for non-STEMI) [11]. Before hospital discharge (minimum stay was 2 days and maximum stay was 8 days, mean = 3 days after PCI) all patients underwent a repeat Doppler echocardiographic examination using the same protocol and measurements were made by a clinician who was blinded to the pre-procedure findings. 

### 2.5. Biochemical Diagnosis of ACS. 

Troponin was measured at admission using Rosh diagnostics highly sensitive troponin (T) with a normal cutoff value of 0.014 ng/dL. CK-MB was also measured using Rosh Diagnostics with a normal range of 1.72–4.78 ng/dL for males and 1.39–3.61 ng/dL for females [12].

### 2.6. Coronary Angiography and PCI

All patients underwent a diagnostic coronary angiogram using the Judkins procedure and an onsite Toshiba lab and the conventional protocol. The culprit lesion was identified and stented. Other lesions were also treated in another setting according to the patient’s hemodynamic status. 

This study conformed with the Helsinki convention guidelines, the protocol was approved by the local Ethics Committee (Approval No. 17/2016) and all patients gave informed consent to participate in the study. 

### 2.7. Statistical Analysis

The data are summarized using frequencies (percentages) for categorical variables and mean ± standard deviation for continuous variables or median interquartile (IRQ) ranges, when appropriate. The differences between the data before and after PCI were compared using the dependent t-test. The correlations were tested with Pearson coefficients. The predictors of delta Global SR were identified with univariate analysis, and multivariate logistic regression was performed using the stepwise method. The receiver operational characteristic (ROC) analyses were performed and the best cut-off value was determined and, at that point, sensitivity and specificity were determined. A significant difference was defined as *p* value <0.05 (2-tailed). The statistical analysis was performed with SPSS Software Package (IBM Corp., Armonk, NY, USA) version 22.0.

## 3. Results

### 3.1. Patients’ Demographics

The patients’ mean age was 55.7 ± 9.4 years, 23% were females, 51% had hypertension where hypertension was defined as office SBP values ≥140 mmHg and/or diastolic BP (DBP) values ≥90 mmHg [13], 40.1% had diabetes, 23.8% dyslipidemia, 63% were smokers and 15% were in class III angina (Appendix A). Fifty-three (66%) were found to have one vessel disease, 12 (15%) had two vessel disease and 15 (19 %) had multivessel disease. Significant (>70%) luminal stenosis in the LAD was found in 33 (63.5%) patients, the LCx in nine (17%) patients and the RCA in 11 (20%) patients. The mean LV EF was 57 ± 10% and 78% had electrocardiographic (ECG) signs of ischemic abnormalities (Appendix A). 

### 3.2. The effect of PCI on Myocardial Function

Following successful PCI, which was defined according to the ACC/AHA as the achievement of <30% residual diameter stenosis of all the treated lesions assessed visually or using QCA, without an in-hospital major adverse cardiac event (death, MI, or repeat coronary revascularization of the target lesion) [14], the LV dimensions and overall EF remained unchanged; however, the segmental, regional and global myocardial function significantly improved. 

The delta regional SR, PSS and ESS of the territories supplied by LAD significantly increased (*p* < 0.05 for all) as did PSSR, PSS and ESS for the territories supplied by LCx and RCA (*p* < 0.05 for all, Table 2). Global LV strain and SR increased (*p* < 0.05 for both). Delta regional SR of the territories correlated with the successful corresponding LAD PCI (r = 0.22, *p* = 0.04), territories of LCx (r = 0.29, *p* = 0.04) and those of the RCA (r = 0.41, *p* < 0.001). Delta regional PSS also correlated with delta ESS, (*p* < 0.05 for all) except delta ESS for LCx and PSS for LAD (*p* > 0.05 for all, Table 3).

A delta regional SR of <−0.02 in the LAD territories had 67% sensitivity and 74% specificity (AUC = 0.72, CI = 0.54 to 0.91, *p* = 0.02) in predicting successful PCI of culprit LAD stenosis. A delta regional SR of <-0.02 was 67% sensitive and 70% specific (AUC = 0.72, CI = 0.56 to 0.88, *p* = 0.02) in predicting LCx PCI. A mean SR of <−0.03 was 73% sensitive and 67% specific (AUC = 0.72, CI = 0.66 to 0.93, *p* = 0.02) in predicting RCA treated stenosis (Figure 1). Delta PSS and ESS change failed to predict successful LAD, LCx and RCA procedures (*p* > 0.05 for all) (Appendix A). 

A delta global SR <−0.35 predicted multivessel disease with 60% sensitivity, 73% specificity (AUC 0.77, CI 0.62 to 0.92, *p* = 0.001) and a delta SR < 0.31 predicted two vessel disease with 75% sensitivity and 63% specificity (AUC 0.72, CI 0.59 to 0.89, *p* = 0.02). The respective values for predicting one vessel disease were delta global SR < 0.29, with 67% sensitivity and 65% specificity (AUC 0.71, CI 0.56 to 0.88, *p* = 0.02) (Figure 2). Delta global PSS and ESS did not predict multivessel disease (*p* > 0.05 for all) (Appendix A).

The global SR was severely reduced before PCI in multivessel disease compared with one/two vessel disease but was not different after PCI (*p* < 0.001, *p* = 0.69 respectively) (Figure 3). Also, we adjusted the parameters of LV myocardial deformation to sex and age but found no significant differences. In addition, we tested for a possible relationship between the LV mass index and the global SR before and after PCI (Appendix A).

### 3.3. The Impact of Site of Arterial Occlusion on Basal vs. Apical LV Function 

Compared with mid/distal occlusion, proximal occlusion resulted in the profound improvement of the basal and apical SR after PCI. The basal delta regional SR for the LAD territories increased more than that with mid/distal occlusion (delta −0.59 vs. −0.21, *p* = 0.04, respectively), as did the apical delta SR (delta −0.16 vs. −0.08, *p* = 0.01, respectively). A similar pattern was seen in the LCx territories, with respective values (delta basal −0.77 vs. −0.41, *p* = 0.03, respectively) and (delta apical −0.50 vs. −0.18, *p* = 0.01) compared with mid/distal occlusion. Likewise, those of the RCA territories (delta basal −0.32 vs −0.16, *p* = 0.04) and apical (delta apical −0.12 vs −0.09, *p* = 0.03, Table 4). 

### 3.4. Comparison of Delta Regional Basal vs. Apical

The delta SR basal and apical regional territories correlated with the subtending artery of the culprit lesion, particularly with the improvement of the basal SR. The delta basal SR for the LAD territories showed more improvement than the apex (r = 0.26, *p* = 0.02 vs. r = 0.22, *p* = 0.04 respectively) as well as for the RCA territories (r = 0.43, *p* = 0.001 vs. r = 0.37, *p* = 0.01), but the LCx territories did not show a significant difference (r = 0.40, *p* = 0.002 vs. r = 0.39, *p* = 0.003, Figure 4, Appendix A). 

### 3.5. STE vs. TDI in Predicting Treated Culprit Lesions

Compared with the SR TDI, the SR STE was more accurate in predicting culprit lesions: a delta regional SR STE <−0.02 was 77% sensitive and 69% specific (AUC 0.75, CI 0.58 to 0.90, *p* = 0.001)), compared with a SR TDI <−0.02 (67% sensitive and 70% specific (AUC 0.72, CI 0.50 to 0.99, *p* = 0.03)), in predicting successful LAD PCI. The delta global SR STE strongly predicted the presence of multivessel disease compared with the delta global SR TDI ((delta global SR STE <−0.33 had 67% sensitivity, 82% specificity (AUC 0.75, CI 0.57 to 0.92, *p* = 0.01) vs. (delta global SR TDI <−0.32 had 62% sensitivity and 78% specificity (AUC 0.72, CI 0.57 to 0.92, *p* = 0.02, Figure 5)).

### 3.6. Predictors of Delta Global SR

In the univariate analysis, hypertension (HTN) (β = −0.294 (−0.313–0.047), *p* = 0.009, smoking β = −0.244 (−0.289–0.015) =0.03, WMA β= −0.317 (−0.284–0.014), *p* = 0.004 and the number of diseased vessels β = −0.256 (−0.188–0.054), *p* = 0.03 predicted the delta global rise of SR. In the multivariate analysis, only HTN β = 0.263 (0.005–3.159) and the number of diseased vessels β = 0.263 (0.005–3.159), *p* = 0.04) independently predicted a delta global rise of SR after PCI (Table 5 and Table 6).

## 4. Discussion

Findings: In a group of patients with ACS found to have significant culprit lesions treated by conventional PCI, the revascularization procedure did not affect the conventional global marker of systolic LV function, i.e., the ejection fraction. It was associated however with a significant improvement in the segmental and regional myocardial function, which are shown be deformation parameters. Irrespective of the branch PCI, the regional strain and SR increased within 3 days of PCI and the delta regional increase correlated with the revascularized artery, with an average sensitivity and specificity of 70%. The highest accuracy of PCI associated with an increased global SR <−35% was seen in patients with multivessel CAD, with an accuracy of 77%. Also, PCI to proximal branch culprit lesions, of the three epicardial arteries, resulted in greater improvement of basal and apical regional deformation. The basal improvement of the deformation function was more profound than the apical region in LAD and RCA PCI but not in LCx PCI. Of note, the STE-based function assessment was more accurate than those obtained from the one third of patients who underwent TDI deformation. Finally, hypertension and multivessel disease were the only independent predictors of recovery of LV function in this group of patients.

Data interpretation: PCI is known to result in early segmental LV improvement of function in patients with disturbed function under controlled conditions, irrespective of the acuity of presentation. Although the conventional LV ejection fraction does not significantly change, the assessment of segmental performance has been shown to improve within hours of successful revascularization [15,16]. Over quarter of a century ago, we have previously shown a significant early increase in LV long axis shortening and lengthening rates, as well as increase in the amplitude of motion using the digitized M-mode technique [17]. These findings were reproduced later when TDI came into practice [18]. Also, the implication of such LV function improvements was reflected on the cavity filling pattern [19]. The most pronounced findings in our patients were those of the delta increase in LV global and regional SR. While myocardial strain reflects systolic function, its rate represents the velocity of strain changes, thus, in a way, reproducing what older techniques showed [20]. However, the unique advantage of speckle tracking technology is its ability to assess individual segment function as well as regional function. Indeed, such an advantage allowed us to assess the response of regional LV function to PCI in individual arterial disease. Also, it permitted us to evaluate the impact of proximal vs mid vessel disease on regional LV function. Indeed, basal LV function proved to be affected significantly by proximal arterial lesions, irrespective of the artery involved. Basal LV function also proved to recover more profoundly compared to the apical region. These findings are supported by the complex myocardial fiber architecture the LV basal region has, with most of it being circumferential while its subendocardial layer remains longitudinally orientated, compared with the apex which is mainly formed by longitudinal fibers [21]. In addition, the multiple septal branches, diagonals, and marginals do contribute to the basal LV region with a significant degree of blood supply compared to the apex, which is vascularized mainly by the peripheral small branches of the LAD and the RCA [20]. We believe that this study is the first to report such an impact of the location of the arterial culprit lesion on regional LV function recovery after PCI. Such effect seems also to be closely related to the number of diseased vessels, with their various blood flow limitations as well as hypertension, with its effect on myocardial thickness and subendocardial ischaemia.

Limitations: Our study has obvious limitations, mainly related to the relatively small sample volume and the potential impact on statistical analysis, despite that the significant level of PCI-related change in segmental and regional LV function suggests that it is robust, irrespective of the sample volume. Although the induced acute ischemia might have also affected our results, this would have only strengthened our conclusion. We did not have longer term data on these patients to assess potential further improvements of LV function, whether segmental or global. Speckle tracking echocardiography remains a research vehicle; its daily use in clinical practice should optimize operators’ skills and the technique’s accuracy in serving such patients.

Clinical implication: PCI of the culprit lesion in acute coronary syndrome is associated with a significant improvement of segmental and regional LV function assessed by myocardial deformation techniques, particularly STE over and above that obtained using tissue Doppler technique. These findings strengthen the clinical application of STE as an accurate technique for monitoring CAD patients, obtaining baseline values and following them up, particularly when they develop symptoms.

## 5. Conclusions

In ACS, the echocardiographic regional myocardial deformation is very accurate in detecting early recovery of LV myocardial function after culprit lesion revascularization. Also, the findings of this study support the current practice regarding the crucial importance of proximal epicardial vessel PCI treatment on LV function compared to more distal lesions. Finally, they also support the fundamental importance of preserving basal LV regional function by optimum revascularization. 

## Figures and Tables

**Figure 1 jcm-09-00024-f001:**
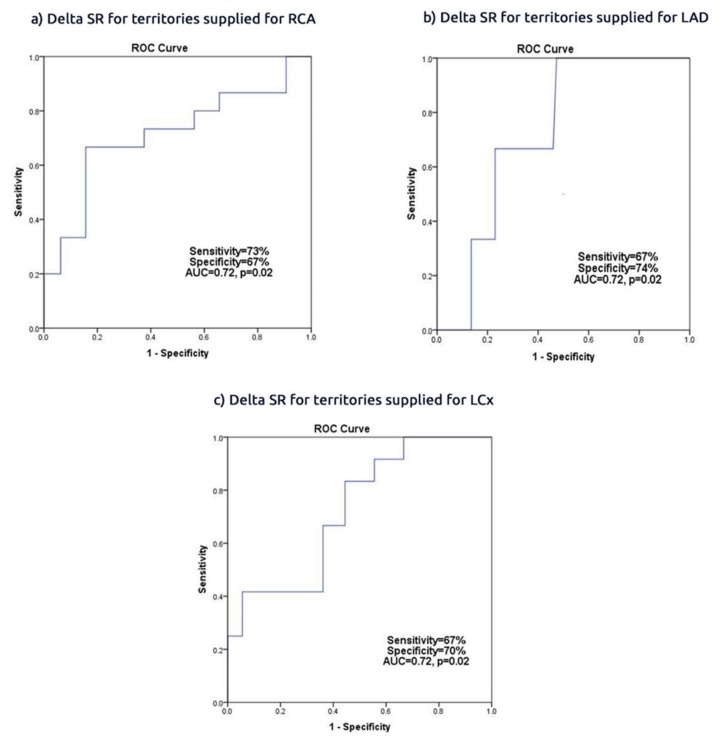
Delta SR in predicting (**a**) territories supplied by RCA; (**b**) territories supplied by LAD; (**c**) territories supplied by LCx.

**Figure 2 jcm-09-00024-f002:**
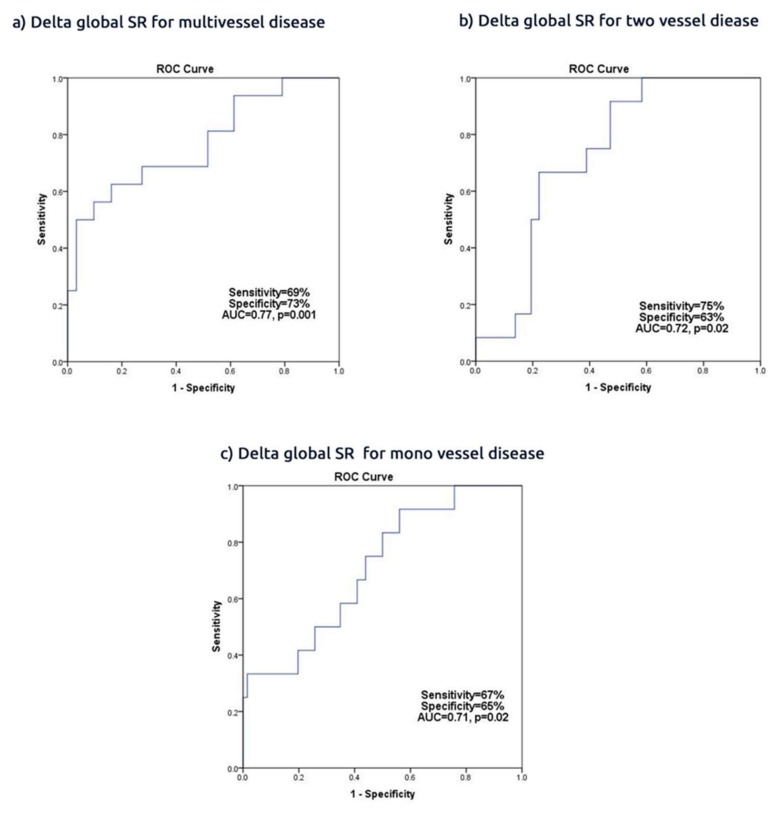
Delta global SR in predicting (**a**) multivessel disease; (**b**) two vessel disease; (**c**) mono vessel disease.

**Figure 3 jcm-09-00024-f003:**
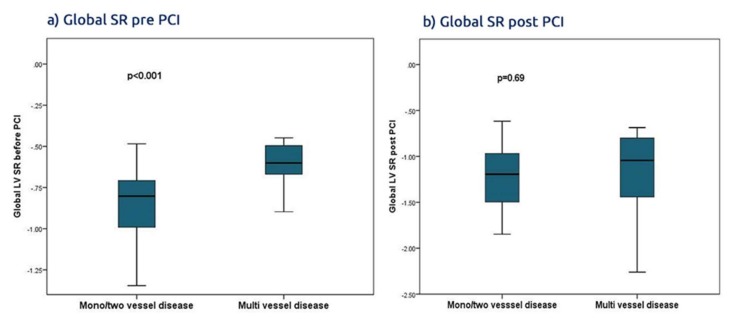
Global SR pre- and post-PCI in multivessel disease vs. mono/two vessel disease.

**Figure 4 jcm-09-00024-f004:**
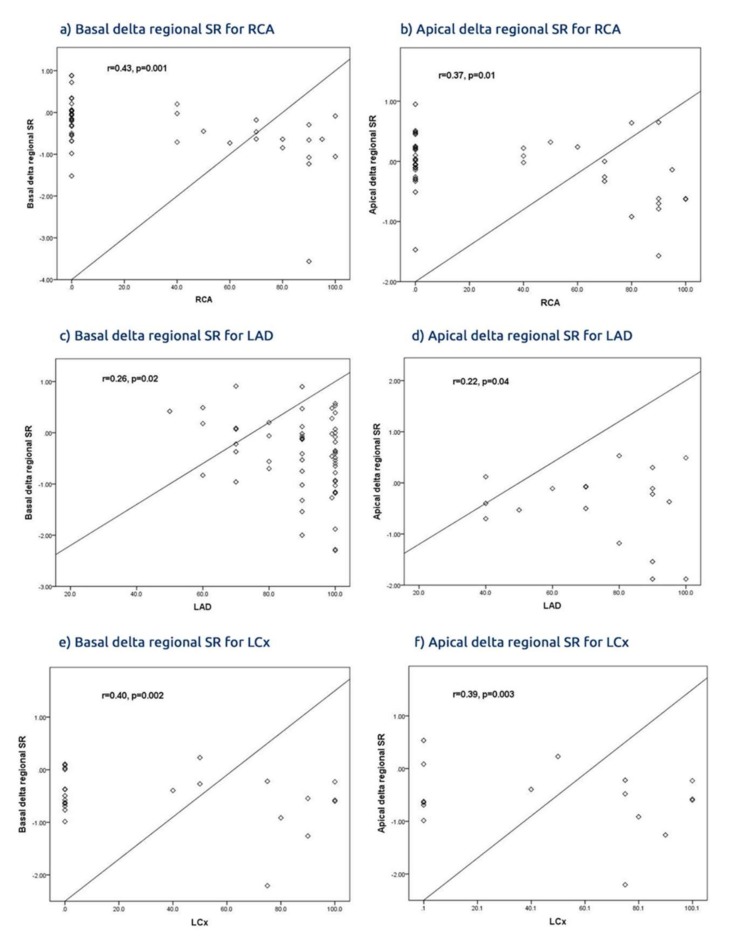
Delta regional of SR supplied for territories arteries; basal vs. apical; (**a**) Delta global speckle tracking echocardiography (STE) SR; (**b**) Delta global tissue Doppler imaging (TDI) SR; (**c**) Delta STE SR for LAD; (**d**) Delta TDI SR for LAD; (**e**) Delta STE R for RCA; (**f**) Delta STE SR for LCx.

**Figure 5 jcm-09-00024-f005:**
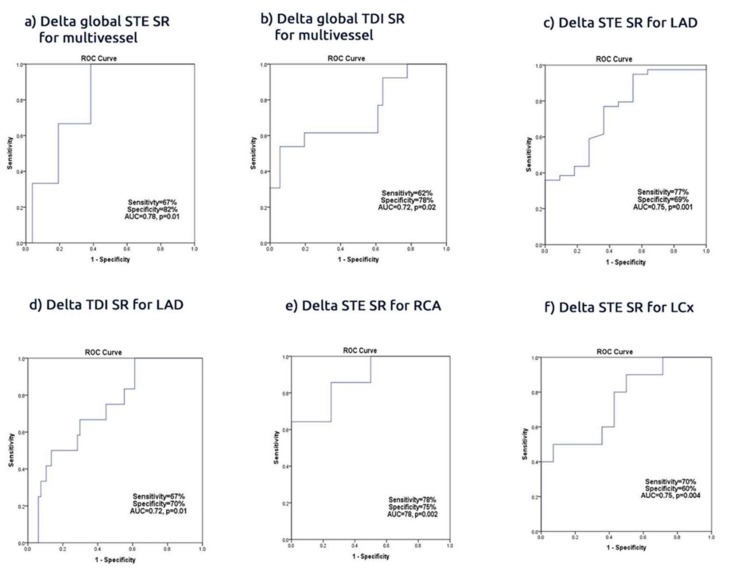
Delta regional and global SR STE vs. TDI. (**a**) Basal delta regional SR for RCA; (**b**) apical delta regional SR for RCA; (**c**) basal delta regional SR for LAD; (**d**) apical delta regional SR for LAD; (**e**) basal delta regional SR for LCx; (**f**) apical delta regional SR for LCx.

**Table 1 jcm-09-00024-t001:** Types of patients with ACS.

Type	STEMI	NSTEMI
N (%)	50 (62.5)	30 (37.5)

ACS: acute coronary syndrome, STEMI: ST segment elevation myocardial infarction, NSTEMI: non-ST segment elevation myocardial infarction

**Table 2 jcm-09-00024-t002:** Strain and Strain rate before and after percutaneous coronary intervention (PCI).

Variable	Before PCI	After PCI	Delta	P value
	(n = 80)	(n = 80)		
**Delta global SR**				
Global systolic strain rate (PSSR)	−0.79 ± 0.20	−1.18 ± 0.34	−0.39	<0.001
Global systolic strain (PSS)	−7.80 ± 2.46	−10.16 ± 3.24	−2.36	<0.001
Global systolic strain (ESS)	−7.97 ± 2.56	−9.05 ± 3.17	−1.08	0.006
**Delta regional SR for LCx**				
Peak systolic strain rate (PSSR)	−0.75 ± 0.30	−0.89 ±0.24	−0.81	0.01
Peak systolic strain (PSS)	−8.0± 4.0	−9.1 ± 3.9	−1.1	0.04
End systolic strain (ESS)	−6.4 ± 3.6	−8.1 ± 3.9	−1.7	0.02
**Delta regional for LAD**				
Peak systolic strain rate (PSSR)	−7.0 ± 0.23	−0.81 ± 0.28	−0.62	0.001
Peak systolic strain (PSS)	−3.6 ± 1.9	−9.6 ± 3.71	−6.0	<0.001
End systolic strain (ESS)	−6.9 ± 2.83	−8.7 ± 3.5	−1.8	<0.001
**Delta regional for RCA**				
Peak systolic strain rate (PSSR)	−0.75 ± 0.3	−0.91 ± 0.36	−0.16	0.049
Peak systolic strain (PSS)	−9.1 ± 4.2	−11.3 ± 4.6	−2.2	0.047
End systolic strain (ESS)	−9.4 ± 4.2	−11.1 ± 4.5	−1.6	0.035

LAD: left anterior descending artery; LCx: left circumflex artery; RCA: right coronary artery; PSSR: peak systolic strain rate; PSS: peak systolic strain: ESS: end systolic strain: SR: strain rate.

**Table 3 jcm-09-00024-t003:** Correlations of deformation parameters (PSSR, PSS, ESS) with coronary disease

Variable	R	P
**Territories supplied by LAD**		
Delta peak systolic strain rate (PSSR)	0.22	0.04
Delta peak systolic strain (PSS)	0.18	0.31
Delta end systolic strain (ESS)	0.21	0.04
**Territories supplied by LCx disease**		
Delta peak systolic strain rate (PSSR)	0.29	0.04
Delta peak systolic strain (PSS)	0.21	0.048
Delta end systolic strain (ESS)	0.16	0.12
**Territories supplied by RCA disease**		
Delta peak systolic strain rate (PSSR)	0.41	0.001
Delta peak systolic strain (PSS)	0.38	0.008
Delta end systolic strain (ESS)	0.27	0.04

Territories supplied by LAD (apico-septal; basal-anterior, mid-anterior, apico-anterior and apical); territories supplied by LCx (basal-septal, mid-septal, basal-lateral, mid-lateral and apico-lateral territories supplied by RCA (basal-inferior, mid-inferior, apico-inferior, basal-posterior and mid-posterior).

**Table 4 jcm-09-00024-t004:** Delta regional SR supplied for territory arteries; basal vs. apical.

Variable	Proximal	Mid/distal	P value
	Occlusion	Occlusion	
	(Delta)	(Delta)	
**LAD**			
Basal SR	−0.59	−0.21	0.04
Apical SR	−0.16	−0.08	0.01
**LCx**			
Basal SR	−0.77	−0.41	0.03
Apical SR	−0.50	−0.18	0.01
**RCA**			
Basal SR	−0.32	−0.16	0.04
Apical SR	−0.12	−0.09	0.03

Regional basal for LAD: Peak systolic SR basal-anterior; apical for LAD: Peak systolic SR apico-septal, Peak systolic SR apico-anterior; Regional basal for Cx: Peak systolic SR basal-septal, Peak systolic SR basal-lateral; Regional apical for Cx: Peak systolic SR apico-lateral; basal for RCA: Peak systolic SR basal-inferior, Peak systolic SR basal-posterior; apical for RCA: Peak systolic apico-inferior.

**Table 5 jcm-09-00024-t005:** Univariate and multivariate predictors of delta Global SR.

Variable	Beta (95% CI)	P	Beta (95% CI)	P
*Univariate predictors*			*Multivariate predictors*	
Age	−0.022 (−0.008–0.071)	0.84		
Gender	0.126 (−0.080−0.276)	0.28		
DM	0.156 (−0.027–0.153)	0.16		
HTN	−0.294 (−0.313–0.047)	**0.009**	0.263 (0.005–3.159)	**0.008**
Dyslipidemia	−0.007 (−0.167–0.157)	0.92		
Smoking	−0.244 (−0.289–0.015)	**0.03**	0.260 (0.011–6.093)	0.115
Angina	0.185 (−0.016–0.169)	0.1		
SBP	0.026 (−0.003–0.004)	0.82		
DBP	−0.057 (−0.009–0.005)	0.61		
HR	0.107 (−0.003–0.008)	0.35		
ECG abnormality	−0.092 (−0.236–0.100)	0.42		
WMA	−0.317 (−0.284–0.014)	**0.004**	1.057 (0.939–1.191)	0.328
No. of vessel disease	−0.256 (−0.188–0.054)	**0.02**	1.050 (0.801–1.377)	**0.04**
LV EDD	−0.007 (−0.004–0.002)	0.58		
LV ESD	−0.073 (−0.016–0.008)	0.52		
LV EF	0.045 (−0.005–0.008)	0.69		
E/e’ ratio	0.912 (1.791–1.050)	0.21		
Aorta	−0.195 (−0.023–0.002)	0.08		
LA diameter	0.029 (−0.124–0.160)	0.8		
Troponin	−0.007 (−0.004–0.002)	0.5		
CK-MB	−0.002 (−0.001–0.002)	0.98		
Creatinine	−0.079 (−0.491–0.236)	0.48		

SBP: systolic blood pressure; DBP: diastolic blood pressure; HR: heart rate; DM: Diabetes mellitus; HTN: Hypertension; ECG: Electrocardiography; CK-MB: Creatine kinase-MB. Bold font reflects significance

**Table 6 jcm-09-00024-t006:** Relationship between LVMI and Global SR.

Variable	R	P value
Before PCI		
Global systolic strain rate (PSSR)	0.02	0.83
Global systolic strain (PSS)	−0.08	0.45
Global systolic strain (ESS)	−0.14	0.20
After PCI		
Global systolic strain rate (PSSR)	0.13	0.24
Global systolic strain (PSS)	−0.07	0.59
Global systolic strain (ESS)	−0.07	0.52

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
