# Peer review of "Early Recovery of Left Ventricular Function After Revascularization in Acute Coronary Syndrome"

_jcm, 2019, doi:10.3390/jcm9010024_

Round 1

Reviewer 1 Report

Shenouda et al. analyzed regional and global strain rates in 80 patients with acute coronary syndrome treated before and after percutaneous intervention. They report that regional strain rates in the area of the treated vessel significantly improved and concluded that the echocardiographic regional myocardial deformation is accurate in detecting early recovery of left ventricular myocardial function after culprit lesion revascularization. I have outlined my suggestions and concerns below:

Major:

- The manuscript has no section on statistics. Hence, one cannot evaluate how the data were analyzed. It is not clear whether absolute or relative differences were used.

- No information is provided in the methods section on how far apart the two measurements took place. There is a sentence in the discussion section stating that the echocardiographies were three days apart. If this were the case a mean and a range for the time difference should be provided.

- Since STEMI and NSTEMI patients were included in this study, a table showing that these two patient populations did not from each other needs to be shown.

- there is no definition of “successful PCI” and hypertension. However, these definitions are critical for the review to assess the value of their findings.

- the unadjusted ROC figures should at least be adjusted for other patient charactistics like age and sex.

Minor:

- sometimes abbreviations are introduced prior to definition (e.g. ln 88 ROI (region of interest)) and sometimes after (e.g. ln 90 peak end systolic strain rate (PESSR)).

Author Response

                         Response to Reviewer Comments

Shenouda et al. analyzed regional and global strain rates in 80 patients with acute coronary syndrome treated before and after percutaneous intervention. They report that regional strain rates in the area of the treated vessel significantly improved and concluded that the echocardiographic regional myocardial deformation is accurate in detecting early recovery of left ventricular myocardial function after culprit lesion revascularization. I have outlined my suggestions and concerns below:

Major:

Point 1.The manuscript has no section on statistics. Hence, one cannot evaluate how the data were analyzed. It is not clear whether absolute or relative differences were used.

Response: We apologize for this mistake. We have now added a section on statistics

Point 2: No information is provided in the methods section on how far apart the two measurements took place. There is a sentence in the discussion section stating that the echocardiographies were three days apart. If this were the case a mean and a range for the time difference should be provided

Response: Thank you! We have now added this information

Point 3: Since STEMI and NSTEMI patients were included in this study, a table showing that these two patient populations did not from each other needs to be shown.

Response: Thank you! We have now added this information in table 1.

Point 4: There is no definition of “successful PCI” and hypertension. However, these definitions are critical for the review to assess the value of their findings.

Response: Thank you! We have added the definitions of PCI and hypertension.

Point 5: The unadjusted ROC figures should at least be adjusted for other patient characteristics like age and sex.

Response: Thank you for this comment. We have adjusted the ROC curve to age and sex. The results did not show any difference.  

Minor:

Point 1: Sometimes abbreviations are introduced prior to definition (e.g. ln 88 ROI (region of interest)) and sometimes after (e.g. ln 90 peak end systolic strain rate (PESSR)).

Response: Thank you! We have corrected this in the revised manuscript.

Best Regards

on behave of the authers

Dr Rafik Shenouda MD

Reviewer 2 Report

Authors have assessed the accuracy of echocardiography in the detection of LV systolic function recovery after an acute coronary syndrome .

Eighty patients (50 % with hypertension, 40 % with diabetes and 23 % with dyslipidemia) with ACS were studied by echocardiography for the evaluation of left ventricle regional wall motion abnormalities, peak systolic strain rate , peak systolic strain and end systolic strain before and after PTCA

The results show an improvement after PTCA, of regional peak systolic strain rate, peak systolic strain and end systolic strain measured in those segments subtended by the culprit vessel. Global strain an strain rate also improved .

Changes in global strain rate were predicted by hypertension and by the number of vessels diseases.

Major comments

This is a nice study , assessing echocardiographic speckle tracking imaging for the detection of LV contractility after acute ischemia treatment

Methods are clearly described , and results are reported in detail

Have authors analysed whether the presence of LVH (increased LV mass index) was related to changes in global strain rate?

Was the time interval between treatment and post procedure echocardiography related to speckle tracking parameters improvement ?

Author Response

                                 Response to reviewer

Major comments

Point 1: This is a nice study, assessing echocardiographic speckle tracking imaging for the detection of LV contractility after acute ischemia treatment. Methods are clearly described, and results are reported in detaiHave authors analyzed whether the presence of LVH (increased LV mass index) was related to changes in global strain rate?

Response: Thank you for this comment. We have now tested the relationship between the LVM index and global strain rate and added the findings in the results section.

Point 2: Was the time interval between treatment and post procedure echocardiography related to speckle tracking parameters improvement?

Response: Thank you . The time of the post PCI echocardiogram was not related to the speckle tracking results, probably because of the short follow up period in a small sample volume 

 best regards

on behave of the authors,

Dr Rafik Shenouda ,MD

Round 2

Reviewer 1 Report

Thanks for the edits.

Reviewer 2 Report

authors have addressed the points raised by this reviewer